# Grading of Intraventricular Hemorrhage and Neurodevelopment in Preterm <29 Weeks’ GA in Canada

**DOI:** 10.3390/children9121948

**Published:** 2022-12-12

**Authors:** Vibhuti Shah, Natasha Musrap, Krishanta Maharaj, Jehier Afifi, Walid El-Naggar, Edmond Kelly, Amit Mukerji, Prakesh Shah, Michael Vincer

**Affiliations:** 1Department of Paediatrics, Mount Sinai Hospital, Toronto, ON M5G 1X5, Canada; 2Department of Paediatrics, University of Toronto, Toronto, ON M5S 1A1, Canada; 3Department of Paediatrics, Dalhousie University, Halifax, NS B3H 4R2, Canada; 4Department of Pediatrics, McMaster University, Hamilton, ON L8S 3Z5, Canada

**Keywords:** infant–newborn, cerebral ultrasound, neurodevelopmental outcome

## Abstract

Objective: The aim of this study was to evaluate the neurodevelopmental outcome at 18–24 months in surviving preterm infants with grades I–IV intraventricular hemorrhages (IVHs) compared to those with no IVH. Study Design: We included preterm survivors <29 weeks’ GA admitted to the Canadian Neonatal Network’s NICUs from April 2009 to September 2011 with follow-up data at 18–24 months in a retrospective cohort study. The neonates were grouped based on the severity of the IVH detected on a cranial ultrasound scan and recorded in the database: no IVH; subependymal hemorrhage or IVH without ventricular dilation (grades I–II); IVH with ventricular dilation (grade III); and persistent parenchymal echogenicity/lucency (grade IV). The primary outcomes of neurodevelopmental impairment (NDI), significant neurodevelopmental impairment (sNDI), and the effect modification by other short-term neonatal morbidities were assessed. Using multivariable regression analysis, the adjusted ORs (AOR) and 95% of the CIs were calculated. Results: 2327 infants were included. The odds of NDI were higher in infants with grades III and IV IVHs (AOR 2.58, 95% CI 1.56, 4.28 and AOR 2.61, 95% CI 1.80, 3.80, respectively) compared to those without IVH. Infants with an IVH grade ≤II had similar outcomes for NDI (AOR 1.08, 95% CI 0.86, 1.35) compared to those without an IVH, but the odds of sNDI were higher (AOR 1.58, 95% CI 1.16, 2.17). Conclusions: There were increased odds of sNDI in infants with grades I–II IVHs, and an increased risk of adverse NDI in infants with grades ≥III IVHs is corroborated with the current literature.

## 1. Introduction

Intraventricular hemorrhage (IVH), an important complication of preterm birth and a major contributor to adverse neurologic outcomes, is inversely related to gestational age (GA) [i.e., the lower the GA, the higher the risk of IVH]. Intraventricular hemorrhages are classified into grades I to IV according to the criteria of Papile et al. [1]. While modifications to this classification exist [2], the application of Papile et al.’s criteria on cranial ultrasound findings continues to be the most commonly used method for diagnosis [3]. Globally, the rate of grades III–IV IVHs is reported to be between 5 and 52%, with marked variations noted between different regions of the world [4].

The grading of IVH is used by clinicians to predict neurodevelopmental outcomes and to counsel families; however, there is no consensus in the literature regarding the impact of grades I–II IVHs and neurodevelopmental outcomes [3,5,6]. Some studies suggest that grades I–II IVHs are benign and have outcomes comparable to those with normal head ultrasound findings [7,8,9]. On the other hand, there are several studies that have reported adverse outcomes with these “minor” hemorrhages [5,10,11,12,13]. Sakaue et al. [12] suggests that low-grade IVH impacts the superior cerebellar peduncles at term-equivalent age magnetic resonance imaging (MRI) which correlates with poor motor development at 3 years of age. Additionally, Vohr and colleagues studied infants who sustained early onset grades I–II IVHs scored lower on the Stanford–Binet intelligence test and had a higher incidence of cerebral palsy compared to those without early-onset hemorrhages at three years of age (corrected age, CA) [10]. Similarly, in a study of 507 surviving infants with a birth weight < 1000 g, infants with grades I–II IVHs had lower mean cognitive scores and significantly worse neurological outcomes than those with normal scans [5].

In contrast, the association between grades III–IV IVHs and adverse outcomes, including an increased risk of handicap [13,14], is well established. It is estimated that 50–75% of infants who have grades III–IV IVHs develop cerebral palsy in childhood, while 45% to 86% suffer major cognitive impairments [15]. Vassilyadi et al. showed that functional independence decreased with higher grades of IVH, and patients with hydrocephalus (ventricular dilatation) fared worse than those without these sequelae [11]. Furthermore, ventriculomegaly following IVH has been recognized as an independent risk factor for adverse cognitive and motor development at 4.5 years of age [16].

While grades III–IV IVHs are major determinants of adverse neurological outcomes, there is no robust consensus regarding the impact of low-grade hemorrhages on neurologic impairment. In a meta-analysis [17], all grades of IVH were associated with higher odds of death or moderate–severe neurodevelopmental impairment (NDI) compared with no IVH; however, a high proportion of the included studies reported unadjusted data, which necessitates cautious interpretation. Given the importance of counseling families on early intervention and management strategies, it is imperative for clinicians to understand the impact of brain lesions (both mild and severe) on long-term outcomes. The aim of this study was to evaluate neurodevelopmental outcomes at 18–24 months CA in a population-representative national cohort of surviving preterm infants born <29 weeks’ GA with normal versus abnormal cranial ultrasound findings, stratified by the severity of IVH.

## 2. Methods

### 2.1. Study Design and Population

Using a retrospective cohort study design, data were collected on all eligible infants born at <29 weeks’ GA and admitted to any of the 28 participating level III neonatal intensive care units (NICUs) in the Canadian Neonatal Network (CNN) between 1 April 2009 and 30 September 2011. The CNN maintains a reliable, standardized national database of perinatal and neonatal information on all NICU admissions, as previously described [18]. Between 18 and 24 months, infants were prospectively followed-up at their respective Canadian Neonatal Follow-Up Network (CNFUN) clinics for a neurodevelopmental assessment. Neonatal and follow-up data of eligible patients in the CNN and CNFUN databases were linked using a unique identifier. We excluded infants with major congenital anomalies, those who were moribund at admission, or had missing information on their IVH diagnoses or follow-up data from the analysis.

### 2.2. Data Extraction/Collection

Information on the outcomes and relevant variables were extracted from medical records by trained data abstractors according to the definitions outlined in the operating manuals [19,20]. Infants were categorized into four study groups based on the grading of ultrasound findings: no IVH, periventricular leukomalacia (PVL), or ventricular dilation (VD) (normal head ultrasound); grades I–II: subependymal hemorrhage or IVH without VD; grade III: IVH with VD (defined as a ventricle size of >10 mm); and grade IV: persistent intraparenchymal lesions (echodense/echolucent), with or without IVH. The categorization corresponds to grades I–IV on Papile’s classification.

### 2.3. Outcome Measures

Surviving preterm infants were assessed for neurodevelopmental outcomes at 18–24 months CA at their respective follow-up clinics, as previously described [21]. Information on hearing impairment and visual acuity was obtained by reviewing the patient’s history or physical examination. Neurological impairment was assessed by administering the Bayley Scales of Infant and Toddler Development III (Bayley-III) by trained examiners with an online certification.

Primary outcome measures included neurodevelopmental impairment (NDI) and significant NDI (sNDI) at 18–24 months CA. Neurodevelopmental impairment was defined as one or more of the following impairments: Bayley-III composite scores < 85 in any of the cognition, language, and motor domains; cerebral palsy with a Gross Motor Function Classification System (GMFCS) score of ≥1; unilateral visual impairment; and/or sensorineural/mixed hearing loss. Significant NDI was marked as the presence of one or more of the following: Bayley-III score < 70 for any of the 3 above components; cerebral palsy with a GMFCS ≥ 3; bilateral visual impairment; or hearing impairment requiring aids/cochlear implants. Secondary outcomes included individual components of NDI. Short-term neonatal morbidities, including bronchopulmonary dysplasia (BPD), defined as a need for oxygen at 36 weeks postmenstrual age (PMA) [22] or at the time of transfer to a level 2 unit, necrotizing enterocolitis (NEC, ≥stage 2 using Bell’s criteria) [23], retinopathy of prematurity (ROP, ≥stage 3, according to international classification) [24], and early-onset (≤48 h of birth) and late-onset (≥48 h of birth) sepsis (defined as positive blood or cerebrospinal fluid culture), were adjusted for amongst the IVH groups.

### 2.4. Statistical Analysis

Maternal and infant demographics and clinical outcomes between the study groups were compared using the Pearson Chi-square test and ANOVA for categorical and continuous data, respectively. The Wilcoxon rank test was used to analyze continuous variables with non-normal distribution. Two models of logistic regression analysis were conducted using grades of IVH as the primary predictor variable with maternal and neonatal factors that were statistically significant in univariate comparisons as covariates. In the first model, odds ratios were adjusted for the center and confounding perinatal factors, including GA, antenatal corticosteroid exposure, sex, small for gestational age (<10th percentile), being outborn, cesarean delivery, maternal hypertension, the Score for Neonatal Acute Physiology-II (SNAP-II), and early onset-sepsis. The second model comprised an assessment of the effect modification by adjusting for the aforementioned factors in addition to the short-term neonatal morbidities, including late-onset sepsis, necrotizing enterocolitis, bronchopulmonary dysplasia, and retinopathy of prematurity. Adjusted odds ratios (AORs) and 95% confidence intervals (CIs) for the outcomes were estimated for each IVH group (grades I–IV) with the ‘no IVH group’ as the reference. A subgroup analysis was carried out to determine whether neurodevelopment in relation to IVH severity followed a similar pattern among GA groups (<26 weeks and 26–28^6/7^ weeks). All statistical analyses were conducted using SAS v.9.3 (SAS Institute, Cary, NC, USA). Two-sided *p*-values of <0.05 denoted a statistical significance.

### 2.5. Ethics Approval and Consent

Approval for data collection was received either from their local research ethics board or quality improvement committees for all participating sites. Parental consent for data collection at a follow-up visit was obtained where required by the research ethics board. This study protocol was approved by Mount Sinai Hospital’s Research Ethics Board (15-0243-C and 7 October 2015).

## 3. Results

Of the 2989 preterm infants admitted to a participating neonatal unit who survived to discharge during the study period, 182 were excluded because of congenital anomalies, and 11 were excluded for missing IVH data (total exclusions, *n* = 193) (Figure 1). Of the remaining infants, 469 (17%) were lost to the follow-up, resulting in 2327 infants included in the analyses. Based on the reported ultrasound findings, 1495 (64.2%) had normal ultrasound findings, 563 (24.2%) had grades I–II IVHs, and 269 (11.6%) displayed grades III–IV IVHs. Table 1 summarizes the neonatal and maternal characteristics according to the IVH grades. Comparisons of the maternal variables revealed differences in maternal hypertension (*p* < 0.01), diabetes (*p* = 0.04), receipt of antenatal steroids (*p* < 0.01), and cesarean delivery (*p* < 0.01) between the groups (Table 1). Differences in neonatal characteristics, including GA, birth weight, sex, outborn status, and severity of illness, were also apparent. A comparison of the major neonatal outcomes revealed significant differences in bronchopulmonary dysplasia, retinopathy of prematurity ≥ stage 3, and late-onset sepsis between the study groups (Table 1) and were predominantly higher among infants with grade IV lesions.

The incidence of NDI at 18–24 months was higher among infants with higher grades of IVH (67.4% grade III and 70.5% grade IV) compared to those with normal or mildly abnormal neurological ultrasound findings (41.9% for those with no IVH and 45.3% for those with grades I–II IVHs). Likewise, rates of sNDI were higher among infants with grades III–IV IVHs (36.8% and 42.2%, respectively) compared to those with no IVH or grades I–II diseases (11.7% and 17.8%, respectively) (Table 2). The incidence of individual components of ND and sNDI are reported in Table 2, with grades III and IV performing worst in all the domains. Infants with IVHs had significantly lower cognitive scores, <85 or <70, and a language score of <70 than those infants with no IVH.

Unadjusted and AORs of the neurodevelopmental outcomes are summarized in Table 3. Compared to infants with no IVH, the odds of NDI and sNDI were higher in infants with grades III and IV hemorrhages (AOR 2.58, 95% CI 1.56, 4.28, and AOR 2.61, and 95% CI 1.80, 3.80) and (AOR 3.58, 95% CI 2.11, 6.07, and AOR 4.76, 95% CI 3.20, 7.11), respectively. There was no statistically significant difference in the NDI between those who had grades I–II IVHs compared to those with no IVH (AOR 1.08, 95% CI 0.86, and 1.35); however, there were higher odds of sNDI for grades I–II IVHs compared to no IVH (AOR 1.58, 95% CI 1.16, and 2.17). An assessment of the effect modification was performed by adjusting for the neonatal morbidities in addition to the perinatal risk factors. The presence of major neonatal morbidities, including necrotizing enterocolitis, bronchopulmonary dysplasia, and retinopathy of prematurity, did not impact the association between the IVHs and neurodevelopmental outcomes (Table 3). There was no statistically significant difference in the NDI/sNDI between those who had grades I–II IVHs compared to those with no IVH for infants < 26 weeks. However, there were higher odds of sNDI for grades I–II IVHs compared to no IVH (AOR 1.60, 95% CI 1.08, and 2.38) for infants between 26–28^6/7^ weeks (Appendix A). 

## 4. Discussion

In this study, we determined that, compared to infants with no IVH, infants with IVH of any grade are at higher odds of sNDI, which increases with IVH severity. Further, the impairment in infants with grades I–II IVHs was mainly in the cognitive and language domains rather than the motor domain or rates of cerebral palsy within this study. Consequently, these results have important implications for patient counseling and potential enrollment in early intervention programs, particularly for infants with mild grades of IVH. It is important to note that extremely preterm infants are predisposed to other underlying conditions and morbidities, such as bronchopulmonary dysplasia, which are also independent predictors of adverse neurodevelopmental outcomes [23]. In our cohort, there was no evidence of an effect modification by short-term neonatal morbidities, such as necrotizing enterocolitis, bronchopulmonary dysplasia, sepsis (early- and late-onset), and retinopathy of prematurity.

Our findings are in accordance with other studies in the literature. In a single center study of extremely low birth weight infants weighing <1000 g, infants born between 1992 and 2000 and diagnosed with grades I–II IVHs exhibited lower Bayley Scales of Infant Development, Mental Development Index scores, and a higher incidence of NDI at 20 months CA [5]. Klebermass–Schrehof et al. also showed that low-grade IVHs were associated with significant neurodevelopmental impairment in preterm infants <32 weeks GA [25]. Similarly, in a larger regional cohort, infants with mild IVHs had a higher risk of moderate–severe neurosensory impairment after an adjustment for small for GA, bronchopulmonary dysplasia, retinopathy of prematurity, and male sex (AOR 1.73, 95% 1.22, and 2.46) at 2–3 years of age [6]. In fact, Hollebrandse et al. evaluated survivors born at <28 weeks GA with different grades of IVHs and matched controls at three different time points (cohort from 1991–1992, 1997, and 2005) for their neurodevelopmental outcomes at 8 years and showed that children with grades I–II IVHs had increased risk of developing motor dysfunction and cerebral palsy [26]. Therefore, evidence suggests that such findings at an early age persist well into childhood and adolescence, particularly when abnormal ultrasounds signify the presence of ischemic white-matter injury [14].

In contrast, several studies suggest that mild IVH is not an independent factor for adverse neurodevelopmental outcomes [8,9]. Variations in the reported outcomes between our study and those reported in the literature could be attributed to a number of factors, including a single-center study, the type of study population (including infants with a higher GA who are at a lower risk of IVH), and sample size. Haslam et al. compared various definitions of sNDI published in the literature on the same cohort and demonstrated that the definitions of sNDIs impact both the incidence and association between the risk factors and sNDI [27]. Consequently, direct comparisons between the studies using different modes of neurodevelopmental assessment do not yield much value.

The postulated mechanism of brain injury in mild IVH is impaired cortical development [5]. Between 10–20 weeks of gestation, the germinal matrix serves as a source of neuronal precursor cells, which then switches to act as a source of glial precursor cells that migrate to the cortical regions around the time of birth for extremely preterm infants. The glial cells produce oligodendroglia; however, the absence of these cells may result in impaired myelination and astrocytic precursor cells crucial for cortical development. There is a suggestion that the occurrence of mild IVH may affect neuronal migration and result in brain injury [1,28].

The merit of classifying intraventricular hemorrhages using a grading system is evident in the ability to predict long-term outcomes and provides clinicians with a systematic approach to counsel families during the neonatal period. However, the correlation between low-grade hemorrhages and long-term development remains a grey area. Despite the growing number of reports that correlate the IVH grade with long-term outcomes, it is often difficult to make cross-study comparisons between the existing studies. Such limitations highlight the need for more standardized methods of assessment and criteria used to define neurodevelopmental impairment. Further longitudinal studies that examine the impact of these cerebral lesions on neurodevelopmental outcomes at school age and early adolescence would be favorable.

Strengths of this study include the large national population of preterm infants, adjusted composite outcomes, taking into account the significant perinatal risk factors and short-term neonatal morbidities, and a high follow-up rate of 84%. However, our study is limited by a number of factors. Due to the retrospective nature of the study, our methodological approach was unable to correlate the degree of impairment with the size of parenchymal hemorrhage/echogenicity. Although ultrasonography is routinely used to detect brain hemorrhages [29], cranial ultrasounds tend to underestimate the degree of brain injury [30] and may lack sensitivity in detecting cerebral white matter injuries, and MRIs may provide additional information and aid in prognostication [14]. It has been postulated that undocumented white matter injuries contribute to the high rates of poor neurodevelopmental outcomes in infants without IVHs. As this is a multicenter study, there may be variations in reporting cranial ultrasound scan findings across sites. In particular, the grading of brain hemorrhages by different radiologists may introduce inter-rater variability. We did not examine maternal education and sociodemographic characteristics, which have been noted as independent predictors of neurodevelopmental outcomes, although the evidence suggests that these factors may have a greater influence in early childhood [31]. Finally, information on whether these infants were enrolled in “early intervention programs”, which have the potential to improve both cognitive and motor outcomes, is not available [32,33].

## 5. Conclusions

In our large, retrospective study, infants with grades I–II IVHs had higher odds of significant neurodevelopmental impairment at 18–24 months’ CA. The risk of adverse neurodevelopmental outcomes was significantly higher in infants with severe IVHs, which is consistent with the previously reported findings. These findings provide further information for clinicians when counseling parents of preterm infants.

## Figures and Tables

**Figure 1 children-09-01948-f001:**
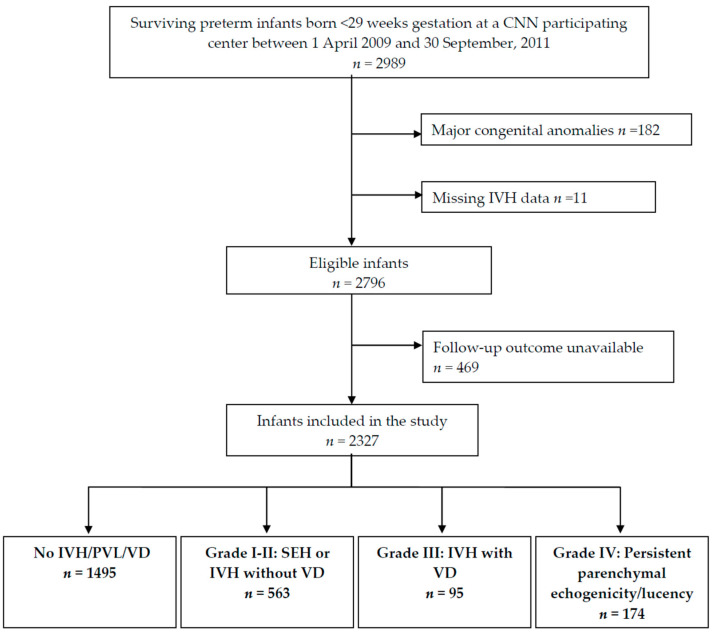
Flow diagram of the study population *. * CNN—Canadian Neonatal Network; IVH—intraventricular hemorrhage; PVL—periventricular leukomalacia; SHE—subependymal hemorrhage; VD—ventricular dilation.

**Table 1 children-09-01948-t001:** Baseline characteristics and clinical outcomes of IVH grades.

Variables *,^1^	No IVH/PVL/VD (*n* = 1495)	Grade I-II: SEH or IVH without VD (*n* = 563)	Grade III: IVH with VD (*n* = 95)	Grade IV: Persistent Intraparenchymal Lesions (Echodense/Echolucent) (*n* = 174)	*p* Value ^2^
**Maternal**					
Maternal age, years	30.9 (5.8)	30.7 (6.1)	30.5 (5.8)	30.9 (6.0)	0.91
Hypertension	287 (19.6)	77 (13.8) *	5 (5.6) *	21 (12.5) *	<0.01
Diabetes	131 (9.2)	31 (5.6) *	6 (6.7)	10 (6.1)	0.04
Antenatal steroids	1341 (91.6)	494 (90.0)	77 (86.5)	140 (82.4) *	<0.01
Rupture of membranes > 24 h	320 (21.8)	107 (19.4)	12 (13.2)	41 (23.8)	0.13
Cesarean section	950 (63.8)	265 (47.2) *	46 (49.5) *	89 (51.5) *	<0.01
**Neonatal**					
Gestational age, weeks	26.6 (1.3)	26.2 (1.4) *	26.0 (1.5) *	25.7 (1.5) *	<0.01
Birth weight, grams	950 (224)	924 (224) *	933 (217)	886 (223) *	<0.01
Male	750 (50.2)	317 (56.3) *	60 (63.2) *	99 (56.9)	0.01
Small for gestational age	NR < 5				
Multiple gestations	435 (29.1)	144 (25.6)	29 (30.5)	49 (28.2)	0.43
Outborn	187 (12.5)	85 (15.1)	22 (23.2) *	30 (17.2)	0.01
Score for neonatal acute physiology (SNAP)-II score ^3^	12 (7, 19)	14 (9, 22) *	21 (14, 30) *	16 (9, 30) *	<0.01
Prophylactic indomethacin	34 (3.2)	21 (4.6)	1 (2.2)	9 (6.0)	0.10
Bronchopulmonary dysplasia ^4^	684 (43.4)	250 (44.4)	43 (45.3)	96 (55.2) *	0.03
Necrotizing enterocolitis ≥ stage 2	99 (6.6)	42 (7.5)	9 (9.6)	16 (9.2)	0.45
Retinopathy of prematurity ≥ stage 3	140 (9.4)	86 (15.3) *	13 (13.7)	35 (20.1) *	<0.01
Late-onset sepsis	394 (26.4)	161 (28.6)	32 (33.7)	67 (38.5) *	<0.01

Abbreviations: IVH—intraventricular hemorrhage; PVL—periventricular leukomalacia; VD—ventricular dilatation. * The denominator varies depending on the availability of the data. ^1^ Results are presented as mean and standard deviation; number, percentage, and median and inter-quartile range, as appropriate; ^2^ significance was assessed across all groups by a Pearson Chi-square test and ANOVA for categorical and continuous data, respectively; ^3^ significance was assessed by a Wilcoxon rank test; ^4^ bronchopulmonary dysplasia among survivors only; * *p* < 0.05 in two-group comparisons with “No IVH” as the reference group. Early-onset sepsis: not reported as the outcome in some cells was <5.

**Table 2 children-09-01948-t002:** Comparison of individual components and composite of neurodevelopmental outcomes among IVH grades.

Variables *,^1^	No IVH/PVL/VD (*n* = 1495)	Grade I–II: SEH or IVH without VD (*n* = 563)	Grade III: IVH with VD (*n* = 95)	Grade IV: Persistent Intraparenchymal Lesions (Echodense/Echolucent) (*n* = 174)	*p* Value ^2^
Cognition score < 85	146 (10.5)	88 (16.9) *	24 (27.9) *	57 (36.8) *	<0.01
Language score <85	443 (32.8)	173 (34.6)	42 (50.0) *	74 (49.3) *	<0.01
Motor score < 85	231 (17.2)	101 (20.4)	31 (37.4) *	85 (57.1) *	<0.01
Any cerebral palsy	41 (2.8)	22 (4.0)	14 (14.7) *	69 (40.8) *	<0.01
NDI	623 (41.9)	254 (45.3)	64 (67.4) *	122 (70.5) *	<0.01
Cognition score < 70	25 (1.8)	18 (3.5) *	9 (10.5) *	19 (12.3) *	<0.01
Language score < 70	110 (8.2)	64 (12.8) *	17 (20.2) *	34 (22.7) *	<0.01
Motor score < 70	51 (3.8)	25 (5.0)	14 (16.9) *	49 (32.9) *	<0.01
sNDI	173 (11.7)	100 (17.8) *	35 (36.8) *	73 (42.2) *	<0.01

Abbreviations: IVH—intraventricular hemorrhage; PVL—periventricular leukomalacia; VD—ventricular dilatation; NDI—neurodevelopmental impairment; sNDI—significant neurodevelopmental impairment. Cerebral palsy (Gross Motor Function Classification System ≥ 3) is not reported as the outcome in some cells was <5. * The denominator varies depending on the availability of the data. ^1^ Only among infants with follow-up data, ^2^ the significance was assessed across all groups by a Pearson Chi-square test and ANOVA for categorical and continuous data; * *p* < 0.05 in two-group comparisons with “No IVH” as the reference group.

**Table 3 children-09-01948-t003:** Comparison of outcomes at 18–24 months corrected age among IVH grades.

Outcome	Level	Unadjusted Odds Ratio (95% CI)	Adjusted Odds Ratio (95% CI) ^1^	Adjusted Odds Ratio (95% CI) ^2^
NDI	Grade I–II vs. No IVH	1.15 (0.94, 1.39)	1.08 (0.86, 1.35)	1.06 (0.85, 1.33)
	Grade III vs. No IVH	2.86 (1.84, 4.44)	2.58 (1.56, 4.28)	2.58 (1.56, 4.27)
	Grade IV vs. No IVH	3.31 (2.35, 4.66)	2.61 (1.80, 3.80)	2.55 (1.75, 3.72)
sNDI	Grade I–II vs. No IVH	1.65 (1.26, 2.15)	1.58 (1.16, 2.17)	1.57 (1.15, 2.15)
	Grade III vs. No IVH	4.42 (2.83, 6.91)	3.58 (2.11, 6.07)	3.66 (2.16, 6.22)
	Grade IV vs. No IVH	5.54 (3.94, 7.79)	4.76 (3.20, 7.11)	4.59 (3.07, 6.87)

Abbreviations: CI—confidence interval; NDI—neurodevelopmental impairment; sNDI—significant neurodevelopmental impairment. ^1^ Adjusted for gestational age, antenatal corticosteroids, sex, small for gestational age, outborn, cesarean delivery, maternal hypertension, Score for Neonatal Acute Physiology-II, early onset sepsis and center. ^2^ Adjusted for gestational age, antenatal corticosteroids, sex, small for gestational age, outborn, cesarean delivery, maternal hypertension, Score for Neonatal Acute Physiology-II, sepsis, necrotizing enterocolitis, bronchopulmonary dysplasia, retinopathy of prematurity, and center. “No IVH” as the reference group.

## Data Availability

We will not be able to provide deidentified individual participant data.

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
