# Peer review of "Grading of Intraventricular Hemorrhage and Neurodevelopment in Preterm <29 Weeks’ GA in Canada"

_children, 2022, doi:10.3390/children9121948_

Round 1

Reviewer 1 Report

This is a very interesting and valuable epidemiological study on IVH in Canada.

What was the minimum number of weeks gestation?

22, 23 or 24 weeks gestation?

How many infants were born at <29 weeks gestation?

Of these, how many died of IVH?

There are a few questions regarding the important factors involved in the development of IVH on the neonatal side.

What was the status of NSAIDS such as acetaminophen, ibuprofen, indomethacin, etc. as prophylactic administration for IVH?

Were sedatives or analgesics administered as prophylaxis for IVH?

How many NSAIDS were used as therapeutic agents for PDA?

Steroids, STA, catecholamines, and pulmonary vasodilators such as NO and PDE III inhibitors, and blood transfusions can also affect circulation.

What was the dosage status with respect to these drugs?

Author Response

Thank you for the positive feedback. Please find below our responses to the issues raised:

1) What was the minimum number of weeks’ gestation? 22, 23 or 24 weeks’ gestation?

The lowest GA of the study population is 22 weeks. We have included preterm infants < 29 weeks who survived to discharge in this study. No changes have been made to the manuscript.

2) How many infants were born at <29 weeks gestation? Of these, how many died of IVH?

The study population of interest is preterm infants <29 weeks GA who survived to discharge and therefore we are not able to answer the question in regards to how many infants were born at <29 weeks’ gestation.

Further, in the CNN database, multiple causes of death may be assigned with one of the reason being IVH, therefore we cannot clearly answer this question.

3) Questions regarding the important factors involved in the development of IVH on the neonatal side.

  1. a) What was the status of NSAIDs such as acetaminophen, ibuprofen, indomethacin, etc as prophylactic administration of IVH?

Information on indomethacin prophylaxis is now included in Table 1. The denominator varies depending on the availability of the data and that has been included in the footnote of Table.1

  1. b) Were sedatives or analgesics administrated as prophylaxis for IVH?

Sedatives or analgesics are not prophylactically used for IVH prevention in the majority NICUs participating in Canadian Neonatal Network.

  1. c) How many NSAIDs were used as therapeutic agents for PDA?

Indomethacin and/or ibuprofen were used as therapeutic agents for medical management of PDA.

  1. d) Steroids, STA, catecholamines, and pulmonary vasodilators such as NO and RDE III inhibitors, and blood transfusions can also affect circulation. What were the dosage status with respect to these drugs?

We agree with the reviewer that postnatal steroids, inotropes, inhaled NO and blood product infusion can affect circulation. However, in our database, we do not have exact doses and timing of administration of these agents in relation to timing of IVH development and thus, unfortunately we cannot make this link.

We expect that our responses to the issues raised are satisfactory. We would like to thank the reviewer for their feedback to improve the quality of our manuscript.

Sincerely,

Dr. Vibhuti Shah, MD, MRCP, FRCPC, MSc

Professor, Departments of Paediatrics and Institute of Health Policy, Management, and Evaluation

Mount Sinai Hospital and University of Toronto

Chair, Research Ethics Board, Mount Sinai Hospital

Reviewer 2 Report

This manuscript describes a large retrospective cohort of preterm infants <29 weeks’ GA and presents relationships between the levels of IVH and developmental outcomes. The manuscript is well written and presents a large cohort of infants born <29 weeks’ GA. The methods are clear and sufficient for reproducibility and statistical analysis is appropriate. There are only minor issue that the authors are encouraged to consider to improve the flow and clarity of the manuscript.

Introduction

The authors present many background studies in the introduction but it is not clear where the gap in literature is identified and what the study in particular will address in terms of bridging the current gap in knowledge.

Assume adverse neurological outcome is inversely related to older gestational age. Please provide a descriptor of the direction of gestational age, but also consider avoiding double-negatives.

Comparison studies and historical findings are discussed in detail in the introduction. Moving this to the discussion may help the flow of the manuscript.

Methods

Clearly articulated participant recruitment and statistical analysis.

Results

Table 1’s legend appears to be absorbed into Table 2’s legend definitions, please provide a separate legend definition for each table.

Tables note that only participants with follow-up data are presented – I assume there is no missing data for each of the variables?

Page 6, line 186: “There was no difference in NDI between those who had grade I-II IVH…” looks like there is a “difference” (probably more accurate to say there is a higher likelihood of NDI – you’ve expressed the result as an Odds Ratio, not a Mean Difference), but there is weak evidence to support a difference in NDI? Within your language of reporting (as opposed to framing associations in terms of strength of association rather than statistical significance), I assume you mean “no significant difference”?

It would be helpful to highlight what the different observation was for the subgroup analysis (page 6, 195) (i.e. that there was little evidence to support an increased likelihood of sNDI in infants <26 weeks). In saying this, it would be interesting to know what the rate of NDI/sNDI was in those under 26 weeks, and those 26-29 weeks.

Discussion

Suggest reviewing and including where appropriate Hollebrandse et al 2021; Klebermass et al 2012; Bolisetty et al 2014

The paragraph beginning page 7, line 241 may be better placed in the introduction

Page 7 line 249-251 is repetitive from the introduction (Page 1 line 40-41)

Page 8 line 272 – The authors could also comment on whether or not data on engagement in early intervention was collected/comment on the implications for not having data available for consideration

Author Response

Thank you for the positive feedback. Please find below our responses to the issues raised:

1) Comment regarding English language: English very difficult to understand/incomprehensive

With due respect to the above comment, we would like to state that the reviewer in the “Comments and Suggestions for Authors
” stated as follows “The manuscript is well written and presents a large cohort of infants born <29 weeks’ GA. The methods are clear and sufficient for reproducibility and statistical analysis is appropriate. There are only minor issue that the authors are encouraged to consider to improve the flow and clarity of the manuscript.”

The manuscript has been revised to improve the readability and corrected for minor grammatical errors.

2) Introduction:  

  1. a) The authors present many background studies in the introduction but it is not clear where the gap in literature is identified and what the study in particular will address in terms of bridging the current gap in knowledge.

The rationale for conducting the study was based on the published studies with unadjusted data, single centre studies and conflicting reports regarding outcomes. The study presented in this study includes data from a population representative national cohort.

  1. b) Assume adverse neurological outcome is inversely related tooldergestational age. Please provide a descriptor of the direction of gestational age, but also consider avoiding double-negatives.

The descriptor of the direction of gestational age is now added. The first sentence in the Introduction section now reads as follows: “Intraventricular (IVH), an important complication of preterm birth and a major contributor to adverse neurologic outcome and is inversely related to gestational age (GA) [i.e..; lower the GA, higher the risk of IVH].”

  1. c) Comparison studies and historical findings are discussed in detail in the introduction. Moving this to the discussion may help the flow of the manuscript.

The paragraph on comparison studies and historical findings are discussed as it sets the stage of why this study was conducted. No changes have been made to the manuscript.

3) Results

  1. a) Table 1’s legend appears to be absorbed into Table 2’s legend definitions, please provide a separate legend definition for each table.

Separate legend definitions have now been included for Table 1 and Table 2.

  1. b) Tables note that only participants with follow-up data are presented – I assume there is no missing data for each of the variables?

Thank you for your comment. There are missing data for each of the outcomes variables and the percentages are calculated from the available data. The denominator can be calculated from the numerator and the percentage presented in the tables.

  1. c) Page 6, line 186: “There was no difference in NDI between those who had grade I-II IVH…” looks like there is a “difference” (probably more accurate to say there is a higher likelihood of NDI – you’ve expressed the result as an Odds Ratio, not a Mean Difference), but there is weak evidence to support a difference in NDI? Within your language of reporting (as opposed to framing associations in terms of strength of association rather than statistical significance), I assume you mean “no significant difference”?

Thank you for your comment. As suggested, the sentence has been rephrased as follows: “There was no difference in NDI between those who had grade I-II IVH compared to those with no IVH (AOR 1.08, 95% CI 0.86, 1.35); however there is a higher odds of sNDI for grade I-II IVH compared to no IVH (AOR 1.58, 95% CI 1.16, 2.17).”

  1. d) It would be helpful to highlight what the different observation was for the subgroup analysis (page 6, 195) (i.e. that there was little evidence to support an increased likelihood of sNDI in infants <26 weeks). In saying this, it would be interesting to know what the rate of NDI/sNDI was in those under 26 weeks, and those 26-29 weeks.

Thank you for your comment. The supplementary files were attached with the manuscript submission and we hope that the reviewers’ had the opportunity to see the results. For infants <26 weeks there was no statistically significant difference in the rate of NDI/sNDI. However, for infants between 26-286/7 weeks, there was higher odds of sNDI for grade I-II IVH compared to no IVH (AOR 1.60, 95% CI 1.08, 2.38). This information is now included in the text on page 6.

4) Discussion

  1. a) Suggest reviewing and including where appropriate Hollebrandse et al 2021; Klebermass et al 2012; Bolisetty et al 2014

The study by Bolisestty et al 2014 has already been included in the “Introduction” section of the manuscript and is reference #5.

The studies by Klebermass et al and Hollebrandse et al 2021 has been cited in the Discussion section, page 7, paragraph 2 as follows:

  1. a) Klebermass-Schrehof et al also showed that low-grade IVH was associated with significant neurodevelopmental impairment in preterm infants <32 weeks GA.25
  2. b) In fact, Hollebrandse et al evaluated survivors born at < 28 weeks GA with different grades of IVH and matched controls at 3 different time points (cohort from 1991-92, 1997 and 2005) for their neurodevelopmental outcome at 8 years and showed that children with grades I-II IVH had increased risk of developing motor dysfunction and cerebral palsy.26
  3. b) The paragraph beginning page 7, line 241 may be better placed in the introduction

Thank you for your comment. The line 241 on page 7 flows with the previous sentence in the paragraph and therefore no changes have been made to the manuscript.

  1. c) Page 7 line 249-251 is repetitive from the introduction (Page 1 line 40-41)

Thank you for your comment. The lines on page 249-251 reads as follows: “There is a suggestion that the occurrence of mild IVH may affect neuronal migration and result in brain injury.1,29 The merit of classifying intraventricular hemorrhages using a grading system is evident in the ability to predict long-term outcomes, and provides clinicians a systematic approach to counsel families during the neonatal period.” While lines 40-41 on page 1 read as follows: “Grading of IVH is used by clinicians to predict neurodevelopmental outcome, and to counsel families; however, there have been conflicting reports in the literature regarding the impact of grade I-II IVH and neurodevelopmental outcome.3,5-6” These lines convey two different messages and so no changes have been made to the manuscript.

We would appreciate input and suggestions from the reviewer if changes are required.

  1. d) Page 8 line 272 – The authors could also comment on whether or not data on engagement in early intervention was collected/comment on the implications for not having data available for consideration

The Canadian Neonatal Follow-up Network does not collect information on whether the infant was enrolled in an “early intervention program” which is associated with improvement in both cognitive and motor outcomes. We have included that as a limitation of the study and have added references 33 and 34 to support the statement.

We expect that our responses to the issues raised are satisfactory. We would like to thank the reviewer for their feedback to improve the quality of our manuscript.

Sincerely,

Dr. Vibhuti Shah, MD, MRCP, FRCPC, MSc

Professor, Departments of Paediatrics and Institute of Health Policy, Management, and Evaluation

Mount Sinai Hospital and University of Toronto

Chair, Research Ethics Board, Mount Sinai Hospital

Reviewer 3 Report

The issue and findings presented in this study is highly relevant to neonatologists and paediatricians across the globe and would be of interest to them. This is a well conducted study. It has significant implications for those counselling parents with preterm infants and children, and for health providers and commissioners.

The incidence of grade I to II IVH appears to be similar to that found in the Epipage 2 study (Chevallier et al 2017), of around 29% (versus 24% in this study). There was only a small amount of missing data (of 11 patients), which makes the data collected very useful.

On Page 3, line 107: the authors stated that NDI is defined as ….. cerebral palsy with GMFCS >/= 1; unilateral visual impairment….. Does this mean that the NDI group and sNDI group are mutually exclusive, and none of the patients in the NDI group are also in the sNDI group? If that is not the case, then surely the definition of NDI needs revising to include any impairment i.e. unilateral and bilateral visual impairment etc. Moreover, would it not be better to classify visual impairment as severe and mild to moderate rather than unilateral or bilateral?

Table 2 – the authors had stated in the table that the variable on the first column is for “Any cerebral palsy”, however on page 6 line 178, they stated that Cerebral Palsy GMFCS >/= 3 was not reported. Please indicate clearly which parameter is being used.

Under discussion, Page 7, lines 228 to 234: the authors suggested that variations in reported outcomes between their study and those reported in the literature could be related to the ongoing improvement in survival rates which may account for similar outcomes of death and adverse neurodevelopmental outcomes. This argument would not explain why those other studies did not find an association between Grade I – II IVH and adverse outcome, unless the authors mean that the increased survival among those of lower gestational age in later studies has affected the overall results of deaths and adverse neurodevelopmental outcomes, and if there are more survivors of extremely preterm infants then this could mean worsening of neurodevelopmental outcome that is not related to the degree of IVH. By comparing baseline characteristics between this study and Wang et al’s study it would appear to be the case (mean gestational age appears lower in this study).This should be made clearer when presenting this argument.

Page 7, lines 234 to 240: the authors suggested that the earlier studies used BSID-II which tends to underestimate development, if that is the case then they would expect more children to be reported to have NDI in those using BSID-II compared to later studies that used Bayley-III. Therefore, I do not feel that this is a valid argument to use for why those studies did not find an association between mild IVH and adverse neurodevelopmental outcomes.

Author Response

Thank you for the positive feedback. Please find below our responses to the issues raised:

1) On Page 3, line 107: the authors stated that NDI is defined as ….. cerebral palsy with GMFCS >/= 1; unilateral visual impairment….. Does this mean that the NDI group and sNDI group are mutually exclusive, and none of the patients in the NDI group are also in the sNDI group? If that is not the case, then surely the definition of NDI needs revising to include any impairment i.e. unilateral and bilateral visual impairment etc. Moreover, would it not be better to classify visual impairment as severe and mild to moderate rather than unilateral or bilateral?

Thank you for your comment. To clarify, the NDI and sNDI group are not mutually exclusive. The definition used for NDI and sNDI are the standard definitions used by the Canadian Neonatal Follow-up Network.  

2) Table 2 – the authors had stated in the table that the variable on the first column is for “Any cerebral palsy”, however on page 6 line 178, they stated that Cerebral Palsy GMFCS >/= 3 was not reported. Please indicate clearly which parameter is being used.

Thank you for the comment. As described on page 3, line 107 NDI includes cerebral palsy with a GMFCS of >1 while sNDI includes cerebral palsy with GMFCS of >3. As stated in the footnote of Table 3 we could provide information on any cerebral palsy of >1. The cell count for cerebral palsy with GMFCS of >3 was <5 and hence could not be included in the table due to possibility of identification of patients.

3) Under discussion, Page 7, lines 228 to 234: the authors suggested that variations in reported outcomes between their study and those reported in the literature could be related to the ongoing improvement in survival rates which may account for similar outcomes of death and adverse neurodevelopmental outcomes. This argument would not explain why those other studies did not find an association between Grade I – II IVH and adverse outcome, unless the authors mean that the increased survival among those of lower gestational age in later studies has affected the overall results of deaths and adverse neurodevelopmental outcomes, and if there are more survivors of extremely preterm infants then this could mean worsening of neurodevelopmental outcome that is not related to the degree of IVH. By comparing baseline characteristics between this study and Wang et al’s study it would appear to be the case (mean gestational age appears lower in this study).This should be made clearer when presenting this argument.

Thank you for your comment. We agree with the reviewers’ concerns and complexity of the issue. Hence we have deleted the sentence in the discussion section.

4) Page 7, lines 234 to 240: the authors suggested that the earlier studies used BSID-II which tends to underestimate development, if that is the case then they would expect more children to be reported to have NDI in those using BSID-II compared to later studies that used Bayley-III. Therefore, I do not feel that this is a valid argument to use for why those studies did not find an association between mild IVH and adverse neurodevelopmental outcomes.

Thank you for your comment. We agree with the reviewers’ concerns and complexity of the issue. Hence we have deleted the sentence in the discussion section.

We have added the following sentence:
Haslam et al compared various definitions of sNDI published in the literature on the same cohort and demonstrated that definitions of sNDI impacts both the incidence and association between risk factors and sNDI.27

We expect that our responses to the issues raised are satisfactory. We would like to thank the reviewer for their feedback to improve the quality of our manuscript.

Sincerely,

Dr. Vibhuti Shah, MD, MRCP, FRCPC, MSc

Professor, Departments of Paediatrics and Institute of Health Policy, Management, and Evaluation

Mount Sinai Hospital and University of Toronto

Chair, Research Ethics Board, Mount Sinai Hospital

Round 2

Reviewer 1 Report

I have no further comment.